# Tolerogenic Dendritic Cell-Based Approaches in Autoimmunity

**DOI:** 10.3390/ijms22168415

**Published:** 2021-08-05

**Authors:** Laura Passeri, Fortunato Marta, Virginia Bassi, Silvia Gregori

**Affiliations:** 1San Raffaele Telethon Institute for Gene Therapy, IRCCS San Raffaele Scientific Institute, 20132 Milan, Italy; passeri.laura@hsr.it (L.P.); fortunato.marta@hsr.it (F.M.); bassi.virginia@hsr.it (V.B.); 2San Raffaele Scientific Institute IRCCS, University Vita-Salute San Raffaele, 20132 Milan, Italy

**Keywords:** dendritic cells, autoimmune diseases, tolerance

## Abstract

Dendritic cells (DCs) dictate the outcomes of tissue-specific immune responses. In the context of autoimmune diseases, DCs instruct T cells to respond to antigens (Ags), including self-Ags, leading to organ damage, or to becoming regulatory T cells (Tregs) promoting and perpetuating immune tolerance. DCs can acquire tolerogenic properties in vitro and in vivo in response to several stimuli, a feature that opens the possibility to generate or to target DCs to restore tolerance in autoimmune settings. We present an overview of the different subsets of human DCs and of the regulatory mechanisms associated with tolerogenic (tol)DC functions. We review the role of DCs in the induction of tissue-specific autoimmunity and the current approaches exploiting tolDC-based therapies or targeting DCs in vivo for the treatment of autoimmune diseases. Finally, we discuss limitations and propose future investigations for improving the knowledge on tolDCs for future clinical assessment to revert and prevent autoimmunity. The continuous expansion of tolDC research areas will lead to improving the understanding of the role that DCs play in the development and treatment of autoimmunity.

## 1. Autoimmune Diseases: Breakdown of Tolerance

Breakdown of immunological tolerance can lead to unwanted and detrimental activation of immune responses against self-antigens (Ags) that causes autoimmune diseases, such as Type 1 Diabetes (T1D), Rheumatoid Arthritis (RA), Multiple Sclerosis (MS), and Inflammatory Bowel Disease (IBD) [1]. These pathologies, usually involving genetic predisposition and poorly defined environmental factors, are widespread. It is estimated that worldwide almost 1 in 10 individuals (7.6–9.4%) suffer from autoimmune diseases [2].

Organ destruction in autoimmune diseases is associated with the dysregulation of the immune system with hyper-reactive effector cells that escape the immune control mediated by the tolerogenic arm of the immune system, leading to hyperactivation of adaptive immune responses and chronic inflammation. Studies in autoimmune patients demonstrated that aberrant activation of immune cells, including lymphoid and myeloid cells, leads to inflammation in the target organs of autoimmunity [3,4,5,6]. Currently, approved therapies for autoimmune diseases involve lifelong administration of immunomodulatory and immunosuppressant drugs that can efficiently improve the outcomes of the disease but, on the other side, are often associated with severe side effects. Specifically, the drugs used in these treatments are non-specific and non-curative, and can induce a systemic, generalized, and persistent immunosuppression leading to the risk of chronic infections or cancer development [7]. The increased knowledge on the relationship among dendritic cells (DCs) and other cells of the adaptive and innate immune system revealed their central role in maintaining tissue homeostasis and tolerance. Further improvements in understanding how DCs promote immunological tolerance and the development of protocols for the manipulation of DC activity in vitro and in vivo lead to actively explore the possibility of identifying effective and specific approaches based on DCs to cure autoimmune diseases. In this context, the use of DCs rendered tolerogenic by different means represents an attractive therapeutic approach for restoring permanent Ag-specific tolerance. Tolerogenic (tol)DCs can be used to specifically target the detrimental immune response against disease associated Ags, while they allow the retention of the capacity of the immune system to be functional and reactive against other pathogens and malignancies [8]. In this review, we briefly introduce the different subsets of human DCs, and we review the involvement of DCs in the induction of tissue-specific autoimmunity. Furthermore, we present current approaches using tolDC-based therapies or targeting DCs in vivo for the treatment of tissue-specific autoimmunity.

## 2. Dendritic Cells Are Central Players in Promoting Immune Responses in Autoimmunity

### 2.1. Human Dendritic Cell Subtypes

DCs are a heterogeneous group of professional antigen-presenting cells (APCs) with the unique ability to induce primary immune responses bridging innate and adaptive immunity. DCs are distributed throughout circulation as well as in lymphoid and non-lymphoid organs. DCs can be classified into conventional (cDCs) and non-conventional DCs, plasmacytoid DCs (pDCs) and monocyte derived DCs (MoDCs) [9]. Human DC subset phenotypes and functions have been extensively reviewed elsewhere; here, we briefly describe their main phenotypical and functional characteristics (Table 1).

Plasmacytoid (p)DCs represent a small subset of cells characterized by the expression of CD123, BDCA-2 (CD303), BDCA4 (CD304) and the lack of CD11c expression [10]. PDCs secrete large amounts of type I IFNs in response to viral infection [11,12] and cross-present Ags to CD8^+^ T cells [13]. More recently, two subsets of pDCs have been described, the CD2^+^ and CD5^+^ pDCs [14,15,16], and single cell RNAseq analysis further refined the pDC classification by including an additional population of cells co-expressing CD2 and CD5 (CD123^+^CD2^+^CD5^+^) and AXL and SIGLEC6. These pDCs do not secrete IFNα and stimulate T cells [17,18] (Table 1).

Conventional (c)DCs, named migratory cDCs, are present in the skin (Langerhans cells), lungs, intestinal tract, liver (interstitial cDCs) and kidneys, while resident or lymphoid cDCs are localized in lymph nodes, spleen, and thymus. cDCs can be classified as cDC1 and cDC2 according to the expression of BDCA3 (CD141) and BDCA1 (CD1c), respectively [9,19] (Table 1). cDC1 cells recognize viral and intracellular Ags and, upon activation, secrete type III IFN and IL-12 [20,21] and present Ags to CD8^+^ T cells, thus they activate cytotoxic T lymphocyte (CTL) responses [21,22,23], and are more active in inducing Th1 responses compared to cDC2 [21]. cDC2 cells recognize, process and present bacterial and exogenous Ags to CD4^+^ T cells, induce the secretion of pro-inflammatory cytokines [24] and promote Th1 and Th17 responses [25,26,27]. More recently, single-cell RNAseq analysis of human peripheral blood defined additional cDC subsets and redefined the existing ones. cDC2 have been redefined as CD1c^+^BTLA^+^ cells [18,28], and the expression of CLEC10A has been proposed to segregate cDC2 into two subclassed DC2A and DC2B, with DC2A being the classical cDC2 and DC2B being a new population of cDC2-related cell subsets [29]. A new population of CD1c^+^ DCs has been reported that express cDC2-associated markers and monocyte-related genes, including S100A8, S100A9, CD14 and CD163 [18]. This population, named DC3, has been defined as CD88^−^CD1c^+^CD163^+^CD14^low^ cells [30,31] (Table 1).

Finally, MoDCs, which mainly differentiate from monocytes in peripheral tissues during inflammation, induce differentiation of CD4^+^ T cells into Th1, Th2 and Th17 cells [32]. MoDCs express markers associated with monocytes and DCs and can be identified as CD14^+^CD1c^+^CD209^+^CD163^+^ cells [16,33] (Table 1).

**Table 1 ijms-22-08415-t001:** Human DC markers and functions.

DC Subsets	Markers	Function	References
**pDC**	CD123	Secretion of type I IFNs Ag cross-presentation to CD8^+^ T cells	[10,11,12,13]
	BDCA-2 (CD303)		
	BDCA-4 (CD304)		
	CD2^high^	IFN-a secretion/T cell stimulation	[14]
	CD2^low^	IFN-a and IL-12p40 secretion/T cell stimulation	
	CD5	No IFN-a secretion	[15]
		Potent B cell stimulator	
	CD2^+^CD5^+^AXL^+^SIGLEC6^+^	No IFN-a secretion	[17]
		Stimulate T cells	
**cDC1**	CD11c	IL12p70	[9,21,22,23,25,26,27]
	BDCA-3 (CD141)	Cross present Ags	
		Th1/Th17 responses	
	CLEC9A	Type III IFN	[20]
		Th1 responses	
**cDC2**	CD11c	Pro-inflammatory cytokines Th1/Th17 responses	[9,24,29]
	BTLA	Reduced IL-12 production, increased TGF b production	[18,28]
		Favor Th2 and FOXP3 polarization	
	CLECA10^+^	More proinflammatory profile than CLEC10^+^	[29]
**DC3**	CD11c	Pro-inflammatory mediators	[18,30,31]
	BDCA-1 (CD1c)	Naive CD4^+^ T cells priming	
	CD14^low^	Th2/Th17 responses	
	CD163		
	CD88^−^		
**Mo-DC**	BDCA-1 (CD1c)	Promote B cell switching	[32,33]
	CD14	Th1/Th2/Th17 responses	
	CD209	Poorly stimulate T cells	

### 2.2. Dendritic Cells and Regulation of Immune Responses

DCs in the immature state (iDCs) predominantly reside in the peripheral tissues and in secondary lymphoid organs [34,35,36] and serve as sentinels of the immune system, continuously patrolling the extracellular milieu. iDCs can recognize a plethora of pathogen-associated molecular patterns (PAMPs) and damage-associated molecular patterns (DAMPs) through the innate pattern-recognition receptors (PRRs), Toll-like receptors (TLRs) or c-type-lectin receptors. iDCs express high levels of PPRs and low levels of major histocompatibility complex (MHC) class II, CD80 and CD86, and their lysosomal activity is attenuated (reviewed in [37]). During classical immune responses, iDCs process the encountered Ags into smaller peptides, which can be presented on the cell surface in the context of MHC class I/II [38,39]. The encountering of the Ag drives the maturation of iDCs that lose their ability to process new peptides and they acquire the capacity to present Ags to T cells. Specifically, DCs upregulate the expression of MHC and co-stimulatory molecules (e.g., CD40, CD80 and CD86), secrete pro-inflammatory cytokines (e.g., IL-1β, IL-12, IL-6, and tumor-necrosis factor α (TNFα)) [40,41] and upregulate the expression of CCR7 and CXCR4, which enable them to migrate to lymph nodes, where they can present Ags to naïve T cells, driving their polarization toward pro-inflammatory Th1, Th2 or Th17 cells or CTLs (reviewed in [42]). For effective activation of T cells, three signals are required: (i) interaction between TCR and Ag/MHC complex; (ii) engagement of CD28 with co-stimulatory molecules (CD80 or CD86); and (iii) secretion of cytokines and chemokines.

Apart from the induction of efficient immune responses against invading pathogens and foreign Ags, DCs are critical modulators of both central and peripheral tolerance. During T cell development in the thymus, DCs play a critical role in the depletion of autoreactive cells. DCs localized in the medulla, together with thymic epithelial cells, present self-Ags to thymocytes, and when the TCR/MHC interaction is strong, they promote self-reactive T cell apoptosis [43]. However, this mechanism does not fully assure the selection of T cell unresponsive to self and innocuous foreign Ags since: (i) self-reactive lymphocytes can escape negative selection; (ii) many innocuous environmental Ags, including commensal microbiota, are not expressed in the thymus; and (iii) TCRs specific for foreign Ags can recognize MHC-self-Ag complexes. To overcome these events, in the periphery are present tolDCs by exploiting several immunosuppressive mechanisms modulate the activity of potentially pathogenic T cells and promote the expansion or/and the differentiation of several subtypes of regulatory T cells, including classical CD4^+^ CD25^hi^ Foxp3^+^ Tregs [44,45,46] and CD49b^+^ LAG-3^+^ type 1 T regulatory (Tr1) cells [47,48].

Four main mechanisms of peripheral tolerance have been described: induction of clonal anergy, metabolic modulation, secretion of anti-inflammatory cytokines and clonal deletion. TolDCs express low co-stimulatory molecules and high inhibitory receptors such as programmed cell death ligand (PDL)-1 [49] and inhibitory Ig-like transcripts (ILTs) [50,51]. These characteristics lead to T cell clonal anergy and T cell unresponsiveness due to Ag presentation in the presence of low co-stimulation, [52], or by the engagement of inhibitory receptors with their ligands expressed on the T cells. The latter include: PDL-1/PDL-2 interaction with programmed death 1 (PD-1) [53,54], the interaction between ILTs and classical and non-classical HLA class I molecules [55,56] and CD80/CD86 binding to the cytotoxic T-lymphocyte-associated protein 4 (CTLA-4). CTLA-4/CD80-CD86 interaction mediates CD80 and CD86 trans-endocytosis and degradation [57] or the induction of indoleamine 2,3-dioxygenase (IDO) [58,59], an enzyme that catabolizes tryptophan, an essential amino acid for T cell proliferation. IDO upregulation in tolDCs leads to: (i) T cell starvation by physical depletion of tryptophan from the local environment [60]; (ii) production of immune-toxic kynurenines that promote T cell apoptosis [61]; and (iii) accumulation of kynurenine, which, upon interaction with the aryl hydrocarbon receptor (AhR) on CD4^+^ T cells, promotes their polarization into Tregs [62]. TolDCs can also alter T cell responses by modulating the metabolic milieu through the expression of heme oxygenase-1 (HO-1), which catabolizes hemoglobin and promotes the production of carbon monoxide, overall reducing DC immunogenicity [63]. Finally, tolDCs, by secreting anti-inflammatory cytokines (i.e., IL-10, TGF-β, and IL-35), are involved in promoting Treg differentiation. TolDC-derived IL-10 suppresses effector T cell responses and induces Tr1 cells [64]. IL-35, which can be secreted by DCs [65], promotes the differentiation of IL-35-producing FOXP3^+^ Tregs and suppress Th17 cell induction [66,67]. Finally, TGF-β promotes the induction of FOXP3^+^ Tregs [40]. In a preclinical model of MS, retinoic acid, a metabolite of vitamin A, has been used to modulate DCs that acquire the ability to induce Tregs and to inhibit Th17 cell polarization [68]. Finally, tolDCs by the expression of FasL and TNF-Related Apoptosis-Inducing Ligand (TRAIL) promote T cell clonal deletion [69,70].

In addition to iDCs, specific DC subsets with tolerogenic properties have been reported. CD103^+^ DCs identified in the intestine are believed to be the major drivers of tolerance in this anatomical district. Indeed, CD103^+^ DCs in the presence of retinoic acid, IDO and TGFβ mediate the differentiation of Tregs and the suppression of Th1 and Th17 cells [71,72]. A population of CD1c^+^ DCs resident in the liver that show an immature-like phenotype characterized by low costimulatory molecules are characterized by the ability to secrete high levels of IL-10 and to induce T cell hypo-responsiveness. Moreover, hepatic CD1c^+^ DCs through an IL-10-dependent mechanism can enhance Th2 polarization and to induce Treg differentiation [73]. CD141^+^ Dermal DCs are immunoregulatory skin resident DCs characterized by the expression of CD141 and CD14 and the constitutive production of IL-10. CD141^+^ Dermal DCs inhibit the proliferation of autoreactive T cells and induce Treg that dampen skin inflammation [74]. Recently, our group identified and characterized a subset of IL-10-producing tolDCs, named DC-10, expressing CD14, CD16, CD141 and CD163. Ex vivo isolated DC-10 have a unique cytokine production profile with high ratio of IL-10/IL-12 production, co-express high levels of the tolerogenic molecules HLA-G and ILT4, prime CD4^+^ T cells, and promote allo-specific Tr1 cells in vitro [75]. CD14^+^cDC2 cells identified in the peripheral blood, ascites and within tumors of cancer patients represent another promising tolerogenic DC subset [76]. CD14^+^ cDC2s express low co-stimulatory molecules, high PD-L1 and transcripts associated with suppression (e.g., osteopontin, COX-2, and thrombospondin-1) and, when co-cultured with naïve CD4^+^ T cells, efficiently inhibit T cell polarization toward pro-inflammatory cells [77,78]. Thus far, CD14^+^ cDC2s have been identified only in the context of cancer, but it seems they have important tolerogenic characteristics. This makes them interesting candidates for future investigations.

The tolerogenic properties of DCs prompt researchers to exploit approaches to generate in vitro or to induce in vivo tolDCs for the treatment of autoimmune diseases (Figure 1).

## 3. Strategies to Generate Ex Vivo Tolerogenic Dendritic Cells

Several approaches to differentiate tolDCs starting from human CD14^+^ monocytes have been established [79]. Although monocyte derived DCs are not superimposable to ex vivo isolated DCs, they recapitulate the properties and functionality of naturally occurring DCs. In vivo iDCs, although tolerogenic, show an intrinsic instable phenotype that can be influenced by the pro-inflammatory environment characteristic of the target organs of autoimmunity. Consequently, protocols to maintain DCs in the immature state were exploited for tolerance induction. In addition, pharmacological agents, immunomodulatory cytokines, exposure to apoptotic cells or genetic manipulation have been used to generate tolDCs in vitro.

### 3.1. Ex vivo Generation of Tolerogenic Dendritic Cells

To keep DCs in the immature state, Machen and colleagues designed a protocol based on the use of specific antisense oligodeoxyribonucleotide (AS-ODN) targeting CD40, CD80 and CD86 transcripts to suppress the protein expression in murine bone-marrow derived DCs [80]. AS-ODN-treated DCs do not secrete nitrogen oxide (NO), TNF-α and IL-12, and a single injection of these cells into prediabetic NOD mice significantly delays the incidence of T1D without affecting the response of T cells to alloAgs. These pre-clinical results led to the translation of AS-ODN to human monocyte-derived DCs and their clinical application [81]. Brown et al. in 2007 proposed an alternative approach to allow Ag presentation by iDCs [82]. Taking advantage of the upregulation of miR155 in murine bone marrow derived DCs upon LPS stimulation, the authors manipulated DCs using lentiviral vectors encoding for GFP and four miRNA155 target sequences, which prevented the expression of GFP by DCs upon LPS activation, thus maintaining the ability of GFP presentation only by iDCs.

Dexamethasone (Dexa), a potent synthetic steroid, has been used to modulate the phenotype of DCs toward a tolerogenic state. Exposure of human CD14^+^ cells, during monocyte-derived DC differentiation, or murine bone-marrow precursors, to Dexa prevents the differentiation of fully maturated DCs, as evidenced by the altered expression of MHC, CD86, CD80 and CD40 molecules, and by the reduced IL-12 production and T cell stimulatory capacity [83]. The active form of Vitamin-D3 (VitD3), 1,25-dihydroxyvitamin D3 (1,25(OH)_2_D_3_), impairs the differentiation of murine and human DCs in vitro and in vivo, leading to a tolerogenic phenotype characterized by low Ag presentation capacity, downregulation of costimulatory molecules, and inhibition of IL-12 production. VitD3-DCs are unable to fully activate T cells and to initiate an immune response [84]. Pre-clinical results in several autoimmune disease models [84] led to the translation of Dexa-DCs and VitD3-DCs into clinical application [85,86,87,88].

Rapamycin (RAPA), a macrocyclic triene antibiotic produced by the bacterium Streptomyces hygroscopicus, with immunosuppressant properties that has been used to generate tolDCs. Addition of RAPA during murine bone marrow derived DC or human monocyte-derived DC differentiation generates cells (RAPA-DCs) with low levels of co-stimulatory molecules (CD86, CD40), high CCR7 expression and a decreased PDL-1 expression [88,89,90,91]. Murine and human RAPA-DCs induce T cell hypo-responsiveness [92] and murine RAPA-DCs expand Foxp3^+^ Tregs in vitro [93]. However, when exposed to LPS murine, RAPA-DCs show a reduced ability to secrete IL-10 and enhancement of IL-12 [94], a cytokine known to be associated with augmented Th1/Th2-polarization of alloreactive CD4^+^ T cells [95].

Two prominent protocols to generate tolDCs in the presence of IL-10 have been established. The first comprises addition of exogenous IL-10 from day 0 during human monocyte-derived DC differentiation (referred to as DC-10) [55]; the second involves the addition of exogenous IL-10 during human monocyte-derived DC maturation (referred to as IL-10DCs) [96]. DC-10, in contrast to IL-10DCs, are fully mature DCs, and express the tolerogenic molecules HLA-G and ILT4, and prime CD4^+^ T cells to become Tr1 cells [56]. On the other hand, IL-10DCs display intermediate levels of CD80, CD83 and CD86 expression, do not produce pro-inflammatory cytokines, express ILT3/4 and promote Treg differentiation [97].

Finally, low concentrations of GM-CSF, a critical growth factor for the generation of murine bone marrow and human monocyte derived DCs, promote the differentiation of tolDCs in vitro [98]. Moreau et al. developed a protocol to generate human monocyte-derived tolDCs, termed autologous tolerogenic DCs (ATDCs) using low-doses of GM-CSF (reviewed in [99]). ATDCs show an immature phenotype being HLA-class II^low^ CD80^low^ CD86^low^ CD40^low^, are semi-resistant to maturation induced by LPS/IFN-γ and secrete IL-10 but no IL-12. Moreover, ATDCs secrete lactate, a factor already described as an immunosuppressive metabolite in the tumor microenvironment [100], which dysregulates the aerobic glycolysis of T cells, resulting in the suppression of T cell proliferation and in the induction of Tregs in vitro [101]. ATCDs have been recently tested in clinical trials to prevent graft rejection after kidney transplantation from living donors [102].

All the evidence demonstrates that tolDCs can be efficiently differentiated in vitro and, independently from the differentiation protocol used, the resulting tolDCs share common features [103]. TolDCs display a semi-mature phenotype expressing low levels of costimulatory molecules, are refractory to maturation induced by LPS or other stimuli, induce T-cell hypo-responsiveness in vitro, secrete immunosuppressive cytokines (e.g., IL-10 or TGF-β) and promote the expansion and/or induction of Tregs.

Multiple evidence from pre-clinical models of autoimmune diseases revealed that tolDC-based approaches are feasible and effective to restore tolerance towards self-Ags; nevertheless, only a limited number of tolDC approaches have been translated into clinical application in autoimmune settings.

### 3.2. Clinical Trial with Ex Vivo-Generated Tolerogenic Dendritic Cells in Autoimmune Diseases

#### 3.2.1. Type 1 Diabetes

T1D is a chronic autoimmune disease resulting in the progressive destruction of β -cells. Insulin replacement therapy is life-saving, but it does not cure T1D patients. Alternative treatments mainly consist of protective therapies, which attempt to regulate immune responses and/or promote β-cell protection/regeneration without tackling the autoimmune pathogenesis [104]. Autoreactive pathogenic T cells recognizing β-cell-derived Ags play a critical role in disease onset and progression [105], while Tregs counteract the activity of pathogenic T cells and promote tolerance [106]. Accordingly, absence or defective Treg functions correlate with autoimmune responses in T1D [107,108]. Therapies targeting pathogenic T cells have been shown to alter the disease course and preserve β-cell mass only in the short term [109,110], providing evidence that restoring the balance between pathogenic T cells and Tregs is not sufficient to cure T1D. To this end, regulatory cell-based approaches, either Tregs or tolDCs, have been proposed for a definitive therapy for T1D patients [111].

In a first-in-man-study, AS-ODN-treated DCs differentiated from monocytes of T1D patients were injected intradermally four times at 2-week intervals in T1D patients (clinicaltrials.gov identifier: NCT00445913) (Figure 2). Treatment was well tolerated and safe, with no observable adverse events or toxicities, and no production of autoantibodies (AAs). Administration of AS-ODN-treated DCs in vivo slightly increased the prevalence of FOXP3^+^ Tregs [81]. In a second trial, autologous mature VitD3-tolDCs loaded with proinsulin peptide C19-A3 were intradermally administered twice in T1D patients (clinicaltrials.gov identifier: NCT04590872). Intradermal injection of proinsulin-epitope-loaded VitD3-tolDCs coincided with low grade toxicity not likely related to the therapy, with no signs of systemic immune suppression, no induction of allergy to insulin, no interference with insulin therapy, and no accelerated loss of β-cell function in patients with the remaining C-peptide. Further studies to investigate whether intradermal administration of autologous proinsulin-epitope loaded VitD3-tolDCs in patients with a shorter diagnosis of T1D and with preserved C-peptide production are currently under testing (Clinical Trial no: NTR5542; Netherlands Trial) [112].

#### 3.2.2. Rheumatoid Arthritis

Rheumatoid arthritis (RA) is an autoimmune disease characterized by chronic synovial inflammation, leading to destruction of joint cartilage and bone. Although the precise etiology remains to be established, it is accepted that RA results from a break in immune tolerance. T cell responses to several joint-associated autoAgs, including citrullinated peptides, can be detected in a portion of RA patients, and Treg functions in peripheral blood are impaired in RA patients with active disease. Immunosuppressive drugs can relieve disease symptoms effectively, but none of the currently available treatments provide a cure (e.g., a long-lasting and drug-free remission of RA) [113]. As an alternative treatment, tolDC-based therapies have been applied to RA patients [114] (Figure 2). Patient-derived monocyte-derived DC generated in the presence of the NF-κB inhibitor (BAY 11-7082) and pulsed with four different citrullinated peptides putative RA autoAgs (Rheumavax) were intradermally injected in RA patients. The treatment was well tolerated, with a 25% reduction in pathogenic T cells and an increase in Treg frequency one month after treatment. Production of IL-15 and inflammatory chemokines was dampened in treated patients and the Disease Activity Scores 28, a common parameter used for the evaluation RA severity, was decreased within a month in treated patients, overall indicating a clinical response [115]. In another study, autologous tolDCs treated with Dexa were intra-articularly injected in knee joints of RA patients. The treatment was safe and well-tolerated and reduced synovitis formation at 3 months after injection was observed (clinicaltrials.gov identifier: NCT03337165) [116]. In an additional clinical trial (AutoDECRA trial; clinicaltrials.gov identifier: NCT01352858), RA patients were treated with tolDCs generated from patients’ monocytes treated with Dexa at day 3 andwith Dexa plus 1,25(OH)_2_D_3_ at day 6 of differentiation, then stimulated with monophosphoryl lipid A (MPLA) to induce a stable semi-mature phenotype, and pulsed with autologous synovial fluid collected from inflamed joints. Resulting tolDCs were injected into the knee joints of RA patients and no worsening knee flares, or other side effects, was observed. Patients treated with the highest dose of tolDCs exhibited an improvement of the clinical symptoms [87]. Another interesting study was carried out in RA patients repetitively injected subcutaneously (around inguinal lymph nodes) with a low or high dose of semi-matured DCs (CreaVax-RA; CRiS identifier: KCT0000035;). CreaVax-RA is composed by autologous semi-mature DCs pulsed with recombinant PAD4, RA33, citrullinated-flaggrin (cit-FLG), and vimentin Ags. Limited side effects were observed, with a significant decrease in AA levels and IFN-γ-secreting T cells. A good-to-moderate EULAR (European League Against Rheumatism) response, which was more pronounced in the DC high-dose group, was observed [117].

#### 3.2.3. Multiple Sclerosis

Multiple sclerosis (MS) is an a chronic neuroinflammatory disease characterized by multifocal demyelinated areas (lesions) caused by immune cell infiltrations across the blood–brain barrier (BBB) that promote inflammation, demyelination, gliosis and neuroaxonal degeneration, resulting in impaired neurological function [118]. Although the pathogenesis of MS remain unknown, myelin-specific T cells are believed to play a crucial role [119], and a defect in Tregs or increased resistance of effector T cells to suppression might contribute to MS induction and persistence [118]. To avoid the general immunosuppression induced by current treatments, Ag-specific immunotherapy is appealing for the treatment of MS [120] and tolDC-based approaches have been proposed. To date, three phase I clinical trials of peptide-loaded tolDCs in MS patients are ongoing or have been recently completed (Figure 2). In a phase Ib clinical trial (clinicaltrials.gov identifier: NCT02283671), eight patients with MS and four with neuromyelitis optica (NMO) received a dose escalation of three intravenous injections of tolDCs generated in presence of Dexa and loaded with a pool of eight MS-related peptides (seven for MS and one for NMO). The treatment was well tolerated, without serious adverse events and with no therapy-related reactions. A significant increase in IL-10 production was observed in PBMCs stimulated with peptides. Moreover, in peripheral blood an increase in Tr1 cell frequency and a decrease in memory CD8^+^ T cells and NK cells compared to the baseline was observed [85]. Two coordinated phase I clinical trials in MS patients (clinicaltrials.gov identifiers: NCT02903537, and NCT02618902) treated with tolDC-VitD3 loaded with a pool of seven myelin peptides are currently ongoing in Belgium and Spain. Beside testing the safety and tolerability of autologous tolDCs-VitD3 pulsed with myelin peptides in a dose-escalation study, the authors aim to compare two routes of tolDC administration, intradermal vs. intranodal injection [120].

#### 3.2.4. Crohn’s Disease

Crohn’s Disease (CD) is a complex chronic inflammatory disease of largely unknown cause. At present, it is believed that CD relies both on genetic predisposition and environmental factors that trigger an aberrant immune reaction against commensal gut flora. Although CD involves defects in both innate and adaptive immunity, it is widely accepted that chronic inflammation is mediated by Th1/Th17 cells [121]. TolDC-based therapy has been proposed to prevent the induction or activation of pathogenic T cells (Figure 2). In a phase I clinical trial aimed to test the feasibility and the safety of a tolDC-based therapy in CD patients (trial number: 2007-003469-42), autologous monocyte-derived DCs were generated in the presence of Dexa and vitamin A and activated with IL-1β, IL-6, TNF-α and prostaglandin E2. The cell product exhibited a semi-mature tolerogenic phenotype with moderated upregulation of CD80, CD86 and CD83, secreted IL-10 in the absence of IL-12 and IL-23 and promoted low T-cell proliferation and IFN-γ production in allogenic mixed lymphocyte reaction (MLR). These tolDCs were administered intraperitoneally in CD patients and treatment was well tolerated, with a clinical improvement observed in one-third of the patients. Interestingly, the number of Th1 and Th17 cells was unchanged in the circulation, but 12 weeks after treatment a significant increase in circulating Tregs was observed. Moreover, T cells from treated patients secreted low levels of IFN-γ upon polyclonal stimulation in vitro, suggesting that tolDCs promoted T cell hypo-responsiveness [122]. Another phase I clinical trial (clinicaltrials.gov identifier: NCT02622763) started enrolling patients with refractory CD to evaluate the safety and clinical efficacy of intralesional administration of autologous Dexa tolDCs. Two different doses of tolDCs will be tested. Thus far, limited information is available because the study was terminated due to low recruitment (three patients).

## 4. In Vivo Dendritic Cell Targeting to Mediate Tolerogenic Responses

The ex vivo generation of tolDCs has some drawbacks, including the extensive manipulation, the fitness of the monocytes isolated from autoimmune patients, and the high manufacturing costs due to tailor-made preparation. To overcome these limitations, new approaches based on in vivo Ag-delivery to naturally occurring tolDCs or Ag co-delivered with immunomodulatory factors to polarize DCs toward a tolerogenic phenotype have been developed.

### 4.1. Nanomedicine to Target Dendritic Cells

In the last decades, evidence demonstrated that Nanoparticles (NPs) were widely used in medicine to deliver drugs or molecules to specific cell subsets [123]. As previously discussed, in the context of autoimmune diseases DCs represent an attractive target for nanomedicine to directly promote their tolerogenic activity in an Ag-specific manner. Indeed, the combined delivery of tolerogenic agents and Ags into NPs promote DCs with the ability to present Ag in a tolerogenic manner [124] (Figure 1). To allow direct phagocytosis from DCs, NPs should have specific characteristics in terms of size, shape, and chemical properties [125]. In T1D, tolDCs presented Ags delivered by NPs encapsulating 1040–55 mimotope peptide and elicited an Ag-specific T cell response in BDC2.5 NOD transgenic mice. In pre-diabetic NOD mice, injection of NPs encapsulating InsB and different tolerogenic agents (e.g., Vit D3 MPs + TGF-β1 MPs + GM-CSF) prevented T1D onset in 40% of the treated mice. Ex vivo analysis revealed decreased levels of insulitis in mice treated with tolerogenic NPs compared to controls, and an increased percentage of FoxP3^+^ Tregs [126]. NPs encapsulating antisense oligonucleotides targeting CD40, CD80 and CD86 molecules injected subcutaneously or intraperitoneally in NOD pre-diabetic mice knocked down the expression of CD40 by 50% and CD86 by 80% on DCs, while slight differences were observed in CD80 expression in DCs. Interestingly, treatment with NPs encapsulating antisense nucleotides with or without InsB_9-23_ in new onset NOD mice reverted diabetes [127]. Finally, intravenous administration of NPs encapsulated with p31Ag, a mimetope recognized by autoreactive T cells in T1D [128], prevented disease development induced by diabetogenic cells in NOD.SCID mice. Histological analysis of the pancreas revealed a normal islet architecture in treated mice compared to controls treated with NPs encapsulated with an unrelated Ag [129].

Although the role of DCs in MS pathology is not crucial, it has been demonstrated that DCs can have a role in the priming of pathogenic T cells and contribute to generating an inflammatory environment in experimental autoimmune encephalomyelitis (EAE) mice [130]. Thus, NPs targeting DCs have been exploited in an EAE model. Maldonado et al. demonstrated that injection of NPs encapsulating PLP, and RAPA inhibited EAE relapse [131]. Moreover, NPs encapsulating IL-10 and MOG injected subcutaneously in mice prior to the induction of EAE showed significant inhibition of EAE development compared to controls. Splenocytes of treated mice, collected at the peak of the diseases, showed a significant impairment in the secretion of IL-17 and IFNγ. Histological analysis on the spinal cord of the treated mice revealed that CD3^+^ infiltrating T cells were increased in a control group compared to mice treated with IL-10/MOG NPs [132].

Celiac Disease (CeD) is an autoimmune disorder of the small intestine, caused by exposure to dietary gluten in genetically susceptible individuals, expressing HLA-DQ2 or HLA-DQ8. The gluten peptides, of which gliadin is the immunodominant one, upon entering intestinal lamina propria are recognized as immunogenic and trigger an adaptive immune response [133]. A potential tolDC-based Ag-specific strategy could modulate gliadin-specific pathogenic T cells and restore gluten tolerance. In a mouse model of CeD, intravenous infusion of gliadin-encapsulating NPs inhibited the proliferation, IFN-γ and IL-17 secretion from gliadin-specific T cells, increased the frequency of FoxP3^+^ Tregs and decreased anti-gliadin antibody production [134]. Based on these promising results, phase I (clinicaltrials.gov identifier: NCT03486990) and phase II (clinicaltrials.gov identifier: NCT03738475) clinical trials have been performed in CeD patients through gliadin-encapsulating NPs (TAK-101). Results showed that intravenous administration of TAK-101 NPs in patients on a gluten-free diet was well tolerated with an acceptable safety profile. In the phase IIa study, the Ag-specific T cell response induced by the gluten challenge was reduced compared to a placebo group, indicating that TAK-101 acts in an Ag-specific manner. Furthermore, TAK-101 NPs led to a reduction of circulating activated CD38^+^ memory T cells, characteristic of CeD-induced intestinal inflammation. TAK-101 NP treatment was also associated with a reduction in gluten challenge-induced intestinal mucosa damages [135].

It is reasonable to believe that NPs might be a suitable approach to restore tolerance in clinical practice. However, further investigation is needed to define the best immunomodulatory molecules to be encapsulated in NPs and the long-term safety profile of the treatment.

### 4.2. Antigen-Delivering Antibodies

An alternative approach to inducing Ag-specific tolerance is the use of Ag-delivering antibodies [136] (Figure 1). This system aims to selectively deliver a particular Ag to DCs using the highly specific binding of monoclonal antibodies (mAbs) to cell surface molecules expressed by naturally occurring tolDCs. The rationale of the approach is based on the hypothesis that, once injected in vivo, mAbs conjugated to an Ag, upon binding to their cognate ligand, are internalized and the delivered Ag is processed and presented by DCs to T cells. Since the purpose is to specifically target DCs with tolerogenic properties, the processed Ag will be presented in a pro-tolerogenic context, leading to the induction/expansion of Tregs, and/or anergy/deletion of Ag-specific effector T cells. Three types of Ag-delivering mAbs have been developed: chemical conjugates between native Abs and Ags; recombinant chimeric Abs; and single-chain fragment variable (scFv) constructs [137]. The first is composed of an Ag chemically conjugated to a native Ab specific for the target molecule and has been demonstrated to be efficient in delivering the Ag to the target cells [138]. The second is a recombinant chimeric Ab that includes the variable regions of mAb specific for the cell surface molecule of choice, and the constant region derived from another immunoglobulin, fused to the relevant Ag. This recombinant chimeric Ab can be genetically modified to optimize the compatibility, solubility, and non-specific binding to Fc receptors [139]. The last construct is composed by a scFv directed to the DCs’ surface molecule linked to the Ag [140].

Due to the pro-tolerogenic properties of DEC-205^+^ BTLA^hi^ DCs [141], the first recombinant chimeric Abs were originally designed to target DCs expressing DEC205 (CD205, LY75) [142]. Pioneer studies on this approach led to the establishment of efficient tolerance in different pre-clinical models of autoimmunity, including EAE, T1D, proteoglycan-induced arthritis (PGIA), and experimental colitis (model for IBD). In EAE, treatment with anti-DEC-205 chimeric Abs fused with MOG_35–55_ or with PLP_139–151_ resulted in amelioration of the disease score, prevention of pathogenic T cell accumulation in the CNS, induction of anergy in T effector cells, and reduction in IL-17 secretion [143,144]. Different hypotheses have been proposed to explain the mechanism through which Ab-mediated Ag delivery to DCs induces this amelioration of EAE symptoms, and, among others, induction of anergy and/or deletion of Ag-specific pathogenic T cells have been proposed [143]. Lately, it was demonstrated that the expansion of pre-existing Tregs plays an important role in this process [145]. Thus, it cannot be excluded that Ag-specific tolerance induction observed in this system is mediated by multiple mechanisms. Several studies in NOD mice demonstrated that anti-DEC-205 chimeric Ab or anti-DCIR2 chimeric Ab fused with insulin or β-cell-derived Ags induced clonal deletion of CD4^+^ and CD8^+^ autoreactive T cells, and conversion of pathogenic CD4^+^ T cells into FoxP3^+^ Tregs (reviewed in [137]). Furthermore, administration of anti-DEC205 coupled with a disease-relevant peptide reduced inflammation and symptom severity in models of PGIA and IBD. At the cellular level, effector T cell deletion/anergy was induced and a portion of the autoreactive CD4^+^ T cells was converted into FoxP3^+^ Tregs [146,147]. An alternative target for Ag delivery to DCs is CLEC9A (DNGR1). Ag-coupled anti-DNGR-1 mAb promoted the proliferation of Ag-specific CD4^+^ T cells and their differentiation into Foxp3^+^ Tregs [148]. Overall, these studies suggest that the Ag-delivering Abs targeting tolDCs might represent an effective therapy for autoimmune diseases. However, the anatomical distribution of DCs makes the delivery of the Ag challenging as much as the induction of tolerance under the pro-inflammatory environment. Thus, further research is necessary to bring this approach to the clinical application.

### 4.3. Epicutaneous Immunotherapy

Epicutaneous immunotherapy (EPIT) is an alternative and novel immune-therapeutic approach to deliver Ag to the APCs localized in the superficial layers of the skin via repetitive applications of an adhesive dermal patch containing a small amount of Ag (Figure 1). In animal models of food allergy, it has been demonstrated that this approach induces desensitization to the given Ag, protects from inflammation and anaphylaxis and induces Tregs (reviewed in [149]). In the skin, Langerhans cells are a subset of DCs that elicit strong immune responses, but also display intrinsic tolerogenic properties in vivo [150]. Uptake of Ag by Langerhans cells plays a central role in the induction of Ag-specific tolerance during EPIT [151]. Several clinical trials of EPIT have recently been completed or are ongoing for pollen, peanuts, and milk allergies with encouraging results in terms of safety and tolerability (reviewed in [149]).

The work on food allergies paved the way for the application of this approach also in autoimmune diseases. In preclinical models of MS, EPIT with MBP was demonstrated to significantly delay the development of EAE both prior to the induction of the disease and upon the development of the initial symptoms. Tolerance induction was mediated by TCRαβ^+^CD4^+^CD8^+^ suppressor T cells, which act through TGFβ [152]. These results led to the development of two in-human studies for EPIT in relapsing remitting MS patients. In the first, patients were treated with an adhesive patch containing a mixture of three immunodominant myelin peptides. The patch was changed once weekly for 4 weeks and then once per month for 11 months (one year in total). This approach led to the activation of Langherans cells at the site of immunization and induced a population of granular DCs in local lymph nodes, characterized by high expression of HLA class II molecules. In the periphery, the treatment promoted Tr1 cells and strongly suppressed myelin-reactive T-cell responses [153]. In a second study, treatment with a myelin peptide skin patch was well tolerated; no serious adverse events were reported with significantly reduced magnetic resonance imaging outcomes (number of Gadolinium^+^ lesions) and clinical symptoms (relapse rate) [154]. The ability of EPIT to induce tolerance was also tested in an animal model of trinitrobenzene sulfonic acid (TNBS)-induced ulcerative colitis. In the study, patches containing TNP-conjugated mouse immunoglobulin (TNP-Ig) induced an amelioration of disease signs accompanied by a reduced production of IFNγ and IL-17 and an increased production of IL-10 from splenocytes [155]. Finally, EPIT protocol was also applied to collagen-induced arthritis (CIA) model. In this case, patches were soaked with type II collagen (COLL II) before CIA induction. Results indicated that EPIT reduces disease severity [156], and similarly to what is seen in the EAE model, epicutaneous application of COLL II at the first signs of CIA results in disease suppression mediated by TCRαβ^+^CD4^+^CD8^+^ suppressor T cells [157].

Given these preclinical and clinical results, it is possible to conclude that EPIT may be an attractive non-invasive therapeutic method for the treatment of different autoimmune diseases. However, more studies are needed to better characterize the mechanism through which this system suppress the pathology and to see whether it is also applicable in more advanced stages of disease.

## 5. Future Prospective

Accumulating knowledge on immune mechanisms at the basis of autoimmune disease has led to the development of many different approaches to generate/induce tolDCs, with the specific aim to restrain, in the long term, unwanted immune responses and restore tolerance. In addition to the promising treatments previously discussed, some new strategies are currently under investigation. DCs can be genetically manipulated with DNA or RNA encoding for Ags, or with DNA encoding for cytokines or co-stimulatory/inhibitory molecules, or using viral vectors including retrovirus, adenovirus, or poxvirus [158]. Viral transduction is an appealing approach for engineering DCs to produce, process and present specific Ags, and to induce the over-expression of immunomodulatory molecules. In particular, the use of lentiviral vectors (LV) has been applied to introduce foreign genes (IL-10, ILT3, and vasoactive intestinal peptide—VIP) into DCs to generate tolDCs [159]. Evidence of the effectiveness of this approach has been produced in a murine model. Specifically, LV-mediated over-expression of IL-10 converted bone marrow-derived DCs into tolDCs that prevented allergic dermatitis in vivo [160] and induced Ag-specific tolerance in an experimental asthma model [161]. Alternatively, MOG35-55-pulsed bone marrow-derived DCs engineered to express VIP prevented EAE development [162]. More recently, LV-encoding ILT3 has been used to generate human tolDCs from CD34^+^ cells. Resulting cells showed decreased expression of co-stimulatory molecules (CD80, CD86), downregulation of NF-κB, low stimulatory activity and promotion of Foxp3+ Tregs in vitro [163]. In addition, LV-mediated IDO, IL-4 or TGFβ gene transfer has been used to generate tolDCs [159]. Our group recently established an efficient protocol to generate IL-10-producing human DCs (DC^IL-^^10^) by genetically engineered monocytes during DC differentiation with a bidirectional LV encoding for IL-10 and a marker gene. DC^IL-^^10^ secrete high levels of IL-10, are phenotypically and functional stable upon exposure to pro-inflammatory signals, and recapitulate the tolerogenic ability of DC-10, as they promote the differentiation of Tr1 cells in vitro and inhibit allogeneic T cell responses in vitro and in vivo [164]. We believe that the combination of IL-10 overexpression with a given Ag will provide engineered DCs with the ability to dampen Ag-specific T cell responses and to restore tolerance via Ag-specific Tregs induction. Preliminary data demonstrated the feasibility of the approach and the ability of Ag-specific engineered murine and human DCs to inhibit Ag-specific T cell responses and to promote Ag-specific Tregs in vitro and in vivo (unpublished data).

Another interesting technology applied in the cancer field is the use of Chimeric Antigen Receptors (CARs), synthetic surface receptors created by the fusion of a single-chain Ag-binding domain derived from an Ab to trans-membrane and intracellular signaling domains [165]. CAR technology has been successfully used to generate highly specific T cells directed against tumor Ags (CAR T cells). Recently, this technology has also been applied to manipulate DCs, and it has been shown that DCs transduced with a CAR directed against CD33 (highly expressed on acute myeloid leukemia (AML) cells) are able to enhance anti-AML CAR T cell cytotoxicity in vitro [166]. More recently, CAR technology has also been applied to monocytes, during macrophage differentiation, leading to the generation of cells able to interact with tumor Ags, and allowing the selective homing of engineered cells to the tumor [167]. Thus, it cannot be excluded that the applications of CAR-DCs might also be used in the field of autoimmune diseases.

## 6. Overall Conclusions

The improved knowledge of tolDCs and the development of protocols to generate cells ex vivo leads to the clinical application of these cells in autoimmune diseases. Overall, the generation of tolDCs ex vivo from patients’ monocytes is feasible and tolDC treatment can be safe and effective for some pathologies. However, there are some limitations that must be considered for improving the development of effective tolDC therapy. In the context of Ag-specific approaches, the selection of immune-relevant Ags is crucial, sometimes challenging or not applicable to all autoimmune diseases due to the lack of associated Ags. To overcome this problem, it has been proposed to pulse T cells with a combination of different disease associated Ags [168]; however, it is unclear if Ag presentation by pulsed tolDCs is stable. Another important aspect to take into consideration is the specific migration of tolDCs to the disease target organ or to the relevant draining lymph nodes to enhance the therapeutic effect. To this end, different routes of administration have been exploited [96], or alternatively, the manipulation of tolDCs by over-expressing specific chemokine receptors to improve tissue-specific homing. Moreover, the stability of the tolDC product upon encountering proinflammatory environments represents an additional important feature; thus, manipulation of tolDC to stabilize their phenotype and functionality might be also considered. All these considerations underlined the need to increase basic knowledge on the biology of tolDCs with the aim of identifying new approaches to improve and/or stabilize their tolerogenic properties. Moreover, the generation of autologous ex vivo tolDCs requires the isolation and differentiation of the cells from patients’ monocytes, which may bear some alterations that will interfere with the functionality of the final cell product [169]. Finally, the production of ex vivo tolDCs requires extreme manipulation, which leads inevitably to high manufacturing costs. As an alternative to ex vivo manipulation, the development of alternative strategies to induce tolerance by autologous tolDCs in vivo can be considered. However, these approaches have been tested primarily in preclinical models of autoimmune diseases; thus, further studies to demonstrate the feasibility, the safety and the efficacy of these new strategies are required for the translation to the clinic.

Gaining knowledge on the biology of monocytes, the starting population used to generate ex vivo tolDCs, as well as on DCs in autoimmune diseases and on tolDCs, will lead to optimize the manufacturing protocols and to identify new approaches for the generation of innovative tolDCs and possible targets for in vivo modulation of DCs in the context of autoimmunity.

## Figures and Tables

**Figure 1 ijms-22-08415-f001:**
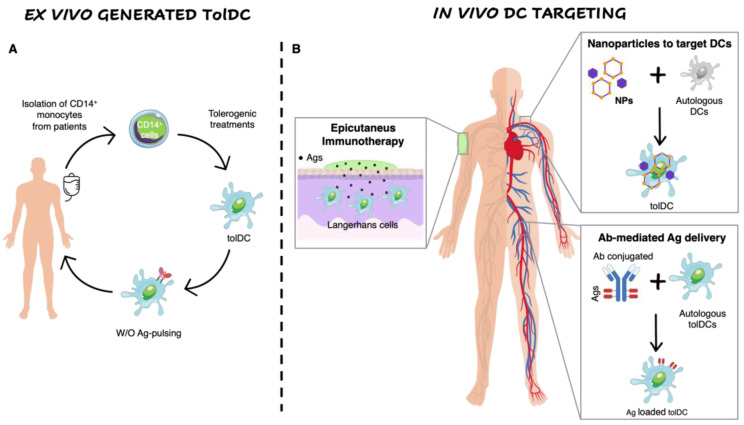
Ex vivo generation of TolDCs vs. in vivo DC targeting. (**A**) Ex vivo generation of TolDCs starts from CD14^+^ monocyte isolation from peripheral blood of autoimmune disease patients. Monocytes are differentiated into DCs in the presence of tolerogenic agents (i.e., Dexa, VitD3, AS-ODN CD80, CD40, CD86) and then pulsed or not with disease-relevant Ags. The obtained cellular product is infused in patients. (**B**) In vivo targeting of DCs. Nanoparticles encapsulating Ags w/o tolerogenic agents target autologous DCs and skew their phenotype toward a tolerogenic one. Antibodies fused to relevant Ags are generated to recognize surface markers expressed specifically by naturally occurring tolDCs (i.e., DEC205, DCIR2, CLEC9A). Epicutaneous immunotherapy that delivers the Ag to the APCs localized in the superficial layers of the skin.

**Figure 2 ijms-22-08415-f002:**
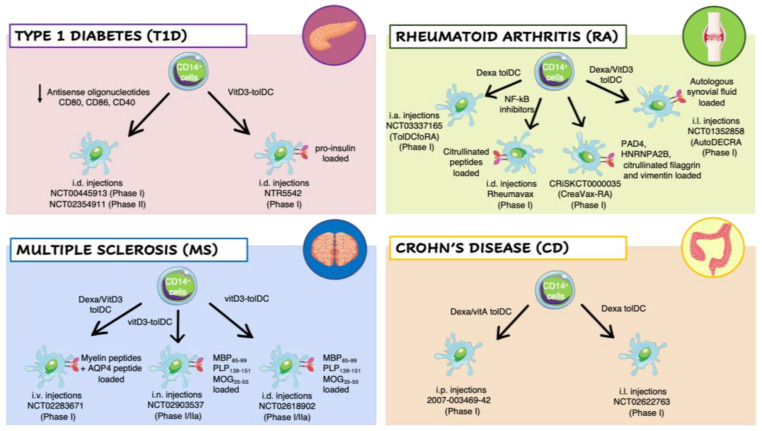
Clinical trials employing ex vivo generated tolerogenic DCs (tolDCs). TolDCs were differentiated starting from patients’ CD14^+^ monocytes in the presence of different tolerogenic agents. The resulting cells have been tested in clinical trials for different autoimmune diseases: Type 1 Diabetes (T1D) (pink panel), Rheumatoid Arthritis (RA) (green panel), Multiple Sclerosis (MS) (blue panel), and Crohn’s Disease (CD) (orange panel). Abbreviations: Dexa, Dexametasone; VitD3, 1,25-dihydroxyvitamin D3; VitA, vitaminA; NF-kB, nuclear factor kappa-light-chain-enhancer of activated B cells; AQP4, acquaporin-4; MBP_85–99_, Myelin Basic Protein; PLP_139–151_, proteolipid protein; MOG_35–55_, myelin oligodendrocyte glycoprotein; PAD4, peptidyilarginine deaminase 4; HNRNPA2B, heterogeneous nuclear ribonucleoprotein A2/B1; i.v., intravenous; i.n., intranodal; i.d., intradermal; i.p., intraperitoneal; i.l., intralesional; i.a., intra-articular.

## Data Availability

Not Applicable.

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
