# Peer review of "Tolerogenic Dendritic Cell-Based Approaches in Autoimmunity"

_ijms, 2021, doi:10.3390/ijms22168415_

Round 1

Reviewer 1 Report

The review by Passeri and colleagues is in general well-written and will be probably interesting for researchers in the DC and autoimmunity field.

However, there are some inaccurancies and other issues that I would like to be considered by the authors to improve their manuscript.

-The most important is that, whenever is possible, they should avoid citing constantly other reviews on the field, instead of the original report. This leads to some errors. For example:

Line 86: Moreover, cDC1 are more active in inducing Th2 responses [16]. Reference 16 is the following review:

[16] M. Haniffa, M. Collin, e F. Ginhoux, «Ontogeny and functional specialization of dendritic cells in human and mouse», Adv Immunol, vol. 120, pagg. 1–49, 2013, doi: 10.1016/B978-0-12-417028-5.00001-6.

I have been reading the section 4.2 of this cited review (Myeloid CD141hi DCs), and there are no references indicating that cDC1 prime Th2 Reponses. So, either the authors cite a proper reference, or they should eliminate this sentence.

Line 88: cDC2 recognize, process, and present bacterial and  exogenous Ags to CD4+ T cells inducing the secretion of pro-inflammatory cytokines 89 [14,17] and primarily promote Th1 responses [16]. All references are again reviews. Again, if we go to reference 16, chapter 4.3. Myeloid CD1c+ DCs, we can read that this DC subset “produce both IL-23 and IL-12p70 is unclear but may imply plasticity for Th1 and Th17 response induction in different contexts”. This does not exactly mean “cDC2 primarily promote Th1 responses”. In fact, CD1c+ DC have high levels of IL-23. In fact, CD1c DCs from obese patienets led to ex vivo Th17 differentiation (Diabetes 61:2238–2247, 2012), and might be involved in autoimmune inflammation (mixed Th1/Th17 responses) (Front Immunol 2017 Aug 17;8:971)

-Line 104: “DCs at immature state (iDCs) predominantly reside in the peripheral tissues”. This sentence may be misleading, iDCs are also present and abundant in secondary lymphoid organs (resident DCs are immature): Blood 102(6):2187-94. J Exp Med (2012) 209 (4): 653–660. Moreover, human spleens contain largely immature cDCs (Immunity46, 504–515, 2017).

In the section 4.2 Antigen-delivering antibodies, it would be nice to include Clec9a (DNGR1) as a potential target. Eur J Immunol. (2010) 40:1255–65. It would be also nice to quote a recent review on this field: (https://doi.org/10.3389/fimmu.2021.643240)

MINOR:

Please, review English grammar and spelling:

-LINE 283 To this end, the USE of different regulatory cell-based

-LINE 302 Several tolDC-based therapies HAVE been tested in patients with RA [14] (Figure 1).

-LINE 330 knee joints of RA patients and no worsening knee flares, or other side effects WERE observed-

-LINE 398 specific characteristics in termS of size, shape, chemical properties [89].

-LINE 435 and IFN-G and IL17 secretion of gliadin

Author Response

Tolerogenic dendritic cell-based approaches in autoimmunity

Manuscript ID: ijms-1269544

Dear Editor,

Dear Referees,

Thank you for giving us the opportunity to resubmit our revised manuscript for publication in IJMS. We thank the reviewers for the criticism and the valuable comments that have contributed to improving the quality of the manuscript. We have modified the paper and inserted the suggestions made by the referees. In the following you find an itemized response to all points raised by the reviewers.

Passeri Laura and Gregori Silvia

List of changes according to comments by referee 1

The review by Passeri and colleagues is in general well-written and will be probably interesting for researchers in the DC and autoimmunity field. However, there are some inaccurancies and other issues that I would like to be considered by the authors to improve their manuscript

-The most important is that, whenever is possible, they should avoid citing constantly other reviews on the field, instead of the original report. This leads to some errors. For example:

Response: We thank the Reviewer for the positive assessment and the suggested modifications for improving our manuscript. We have revised the manuscript by including original manuscripts.

Line 86: Moreover, cDC1 are more active in inducing Th2 responses [16]. Reference 16 is the following review: [16] M. Haniffa, M. Collin, e F. Ginhoux, «Ontogeny and functional specialization of dendritic cells in human and mouse», Adv Immunol, vol. 120, pagg. 1–49, 2013, doi: 10.1016/B978-0-12-417028-5.00001-6. I have been reading the section 4.2 of this cited review (Myeloid CD141hi DCs), and there are no references indicating that cDC1 prime Th2 Reponses. So, either the authors cite a proper reference, or they should eliminate this sentence.

Line 88: cDC2 recognize, process, and present bacterial and  exogenous Ags to CD4+ T cells inducing the secretion of pro-inflammatory cytokines 89 [14,17] and primarily promote Th1 responses [16]. All references are again reviews. Again, if we go to reference 16, chapter 4.3. Myeloid CD1c+ DCs, we can read that this DC subset “produce both IL-23 and IL-12p70 is unclear but may imply plasticity for Th1 and Th17 response induction in different contexts”. This does not exactly mean “cDC2 primarily promote Th1 responses”. In fact, CD1c+ DC have high levels of IL-23. In fact, CD1c DCs from obese patienets led to ex vivo Th17 differentiation (Diabetes 61:2238–2247, 2012), and might be involved in autoimmune inflammation (mixed Th1/Th17 responses) (Front Immunol 2017 Aug 17;8:971)

Response: We apologize for the inaccuracy. We have modified the manuscript accordingly.

-Line 104: “DCs at immature state (iDCs) predominantly reside in the peripheral tissues”. This sentence may be misleading, iDCs are also present and abundant in secondary lymphoid organs (resident DCs are immature): Blood 102(6):2187-94. J Exp Med (2012) 209 (4): 653–660. Moreover, human spleens contain largely immature cDCs (Immunity46, 504–515, 2017).

Response: We revised the concept of immature DCs as suggested by the Reviewer. We included the suggested manuscript.

In the section 4.2 Antigen-delivering antibodies, it would be nice to include Clec9a (DNGR1) as a potential target. Eur J Immunol. (2010) 40:1255–65. It would be also nice to quote a recent review on this field: (https://doi.org/10.3389/fimmu.2021.643240)

Response: We thank the Reviewer for the suggestions and we modified the paragraph by including the reference indicated and wec quoted the review.

MINOR:

Please, review English grammar and spelling:

-LINE 283 To this end, the USE of different regulatory cell-based

-LINE 302 Several tolDC-based therapies HAVE been tested in patients with RA [14] (Figure 1).

-LINE 330 knee joints of RA patients and no worsening knee flares, or other side effects WERE observed-

-LINE 398 specific characteristics in termS of size, shape, chemical properties [89].

-LINE 435 and IFN-G and IL17 secretion of gliadin

Response: The paper was completely double checked for grammatical errors and typos. You can find these changes throughout the paper, in track changes.

Reviewer 2 Report

See attached 

Author Response

Tolerogenic dendritic cell-based approaches in autoimmunity

Manuscript ID: ijms-1269544

Dear Editor,

Dear Referees,

Thank you for giving us the opportunity to resubmit our revised manuscript for publication in IJMS. We thank the reviewers for the criticism and the comments that have contributed to improving the quality of the manuscript. We have modified the paper and inserted the suggestions made by the referees. In the following you find an itemized response to all points raised by the reviewers.

Passeri Laura and Gregori Silvia

List of changes according to comments by referee 2

In this manuscript, Dr. Passeri and colleagues provide an extensive review of dendritic cells (DC) in immune responses and how, over the recent years, this cell population has become a focus for negatively regulating immune responses, especially autoimmunity. The concept of DC-mediated down-regulation of autoimmunity can be traced back two decades to studies carried out by Drs. Strominger, Clare-Salzler and Wilson using the type 1 diabetic NOD mouse model. In their studies, the investigators identified a role for iNKT induction of tolerogenic myeloid DC. This overall concept has slowly emerged, and the current manuscript reviews the current status.

In reviewing the recent research and clinical applications of tolerogenic DCs, the authors present cover four topics: (1) an overview of DC immune functions, (2) strategies to induce tolerogenic DC populations, (3) how these populations are being applied to several autoimmune/inflammatory diseases, and (4) ex vivo strategies to target the interplay between DC abd DC activating cell types. Overall, the review provides an extensive update of this particular field that should be of interest to multiple fields of immunity.

Response: We thank the Reviewer for the positive assessment of our manuscript.

Critique:

(a) While the manuscript is presumably an in-depth, expansive overview of manipulated tolerogenic DC populations and their current/future clinical application, this focus is lost by an extensive introduction of DC biology that greatly detracts from the main theme. Would it not be sufficient to replace the 3 pages (lines 53-196) with a single, simple introductory paragraph and a Table listing the different DC populations and their identification profiles and references? Afterall, there are many aspects of DC that are not covered in the presented description.

Response: We thank the Reviewer for the suggestion. We believe that an introduction of DCs is required to allow the reader to have a general background. Nevertheless, as suggested we shortened this part and we included a summary table (Table 1) including human DC subset phenotype, characteristics and relevant references.

(b) Overall, the manuscript is well-written, but there are numerous minor grammatical errors throughout the text that will need to be corrected.

Response: We revised the manuscript and corrected the minor grammatical errors.

(c) In the first sentence of the manuscript (line 26), there is a reference to allergies being a breakdown in immunological tolerance! Is a hyper-response to an environmental antigen an immune tolerant state?

Response: Since the manuscript is mainly focused on autoimmunity, we eliminated the reference to allergy.

(d) Within Section 3, there are several references to ¨see Clinical trial section¨ or reference to Clinical protocols. These need to be described, referenced or removed.

Response: According to the Reviewer’s request we removed the “see clinical trial section”, and we quoted the original manuscript also in Section 3.

(e) In terms of presentation, it seems logical that Figure 2 should precede Figure 1

Response: We switched the Figures’ order according to the Reviewer’s suggestion.

Reviewer 3 Report

In the present review Passeri et al. describe dendritic cells as regulators of immune response (briefly, but it represents one-third of overall text), and tolerogenic dendritic cells as a tool to struggle against autoimmune diseases. The topic of a review is of current interest.

However, for a review, there are quite a lot of references to other reviews. There are statements that should be better supported by experimental article references, but not reviews, and statements that can be supported by review references, but that are without references at all. At a glance, some plagiarism could be visible with no correct references given (as one example, if you will compare lines 106-108 and the following article: DOI: 10.1016/j.it.2020.11.001, beginning of the paragraph "Immunogenesis versus Tolerogenesis: A Role for Dendritic Cells in Peripheral Tolerance").

Little originality can be found and it is not clear how this review can contribute to the field, except for descriptive values of what already was published in other reviews, or "references list" value. Considering all the mistakes, missing references, confusion due to language, definitions, and in general, the current manuscript seems very immature and requires at least major revision.

Below the most visible critical notes are listed.

1. The following statements should be supported with references:

  • Lines 31-32, “Organ destruction in AIDs is associated to the dysregulation of the immune system, leading to chronic inflammation and hyperactivation of adaptive immune responses.”
  • Lines 68-70, "Once they completed maturation in the BM, pDCs migrate into the bloodstream and are characterized by the ability to secrete large amounts of type I IFNs in response to viral infection."
  • Lines 112-116, "The encountering of the Ag drives the maturation of the iDCs that lose the ability to process new peptides and acquire the capacity to present Ags to T cells. Specifically, they upregulate the expression of MHC and co-stimulatory molecules (i.e., CD40, CD80, and CD86) and secrete pro-inflammatory cytokines (i.e., IL-1β, IL-12, IL-6, and tumor-necrosis factor α (TNFα))."
  • Lines 129-133, "However, this mechanism does not fully assure the selection of T cell unresponsive to self and innocuous foreign Ags since: (i) self-reactive lymphocytes can escape negative selection, (ii) many innocuous environmental Ags, including commensal microbiota, are not expressed in the thymus, and (iii) TCR specific for foreign Ags can recognize MHC-self-Ag complexes."
  • Lines 139-141, "Four main mechanisms of peripheral tolerance have been described: induction of clonal anergy, metabolic modulation, secretion of anti-inflammatory cytokines, and clonal deletion."
  • Lines 149-150, "T cell anergy can be also induced by ... the induction of indoleamine 2,3-dioxygenase (IDO)."
  • Lines 158-161, "In addition to the above mechanisms, tolDCs by secreting immunomodulatory mediators as anti-inflammatory cytokines (i.e., IL-10, TGF-β, and IL-35) and metabolites (i.e., retinoic acid (RA)) are involved in promoting Treg differentiation."
  • Lines 162-164, "IL-35 derived from tolDCs promotes the differentiation of IL-35-producing FOXP3+ Tregs and suppressed Th17 cell induction."
  • Lines 237-239, “Addition of RAPA during DC differentiation generates cells 237 (RAPA-DCs) with low levels of co-stimulatory molecules (CD86, CD40), high CCR7 ex-238 pression, and a decrease PDL-1 expression.”
  • Lines 239-240, “RAPA-DCs induce T-cell hypo-responsive-239 ness and expand Foxp3+ Tregs in vitro.”
  • Lines 421-427, “In the same autoimmune setting, IL-10/MOG encapsulating NPs injected subcutaneously in mice prior the induction of EAE showed significant inhibition of EAE development compared to controls. Splenocytes of treated mice, collected at the peak of the diseases, showed a significant impairment in the secretion of IL-17 and IFN compared to splenocyte of control mice. Histological analysis on the spinal cord of the treated mice revealed that CD3+ infiltrating T cells were increased in control group compared to mice treated with IL-10/MOG NPs.”
  • Lines 428-432, “Celiac Disease (CeD) is an autoimmune disorder of the small intestine, caused by the exposure to dietary gluten in genetically susceptible individuals, expressing HLA-DQ2 or HLA-DQ8. The gluten peptides, of which gliadin is the immunodominant one, upon entering intestinal lamina propria are recognized as immunogenic and triggers an adaptive immune response.”
  • Lines 469-471, “This conjugate has been demonstrated to be efficient in the delivering the Ag. However, it showed a lower target specificity compare to the recombinant chimeric Ab, which present some modification that enhanced their specificity.”
  • Lines 586-588, “In 2016 the CAR technology has been also applied to monocytes leading to the generation of cells able to interact with tumour Ags, allowing the selective homing of engineered cells to tumour.”

2. Line 22: The title of the paragraph does not match the content of the paragraph. It is stated about "breakdown of tolerance" and "dysregulation of the immune system", but the reasons why it is dysfunctional are not discussed. I would suggest either add an appropriate brief discussion on why is it considered dysfunctional, or change the title.

3. Line 27, “(Goodnow et al., 2005).”: Reference is given in a different from the main list style.

4. Lines 62-65, 99-102: It is quite controversial if so-called monocyte-derived dendritic cells can be classified as dendritic cells since they share the phenotype of both dendritic cells and monocytes/macrophages. Moreover, the term "monocyte-derived dendritic cells" is attributed to artificially obtained cells after induction of monocytes with IL-4 and GM-CSF in vitro, which cannot be obtained physiologically. The term "macrophages-DC progenitor" is not clear as well (line 59). Probably "monocytes-dendritic cells progenitor" was meant? Also, following the article referenced in the current manuscript (10.1038/nri3712), monocyte-derived cells are not descendants of a common dendritic cell precursor. Overall, the whole paragraph (lines 55-65) is suggested to be revised, with more references to be added.

5. Lines 54, 103: The numbers of paragraphs are the same.

6. Lines 146-147: "…PDL-1 and PDL-2 programmed death 1 (PD-1) interaction [33]…":

The meaning of this sentence is unclear, revision is required. Also, the reference seems misleading or out of context.

7. Lines 147-148: "...ILTs HLA class I molecules interaction...":

The same as in the previous note - interaction between what and what? There is no hyphens, no prepositions. Language editing is required, otherwise, it just looks like a set of words with no logic.

8. While describing the effects of tolerogenic dendritic cells reported in experiments it is important to add which cell subtypes and their species origin were used to perform experiments in referenced articles.

9. Line 164-166, "In a preclinical model of MS, retinoic acid (RA), a metabolite of vitamin A has been used to induced Tregs and inhibit Th17 cell polarization [41].":

It is clearly written in the present manuscript and was done in the referenced paper that artificial treatment with retinoic acid induced Tregs and inhibited Th17 polarization, but not retinoic acid-secreting dendritic cells. Considering that no reference was given earlier to support that dendritic cells can secrete retinoic acid (see comment 1), it can't be claimed that dendritic cells can act through such a mechanism on T cells.

10. Lines 168-195: It is a debatable question whether dendritic cells can express CD14, or if we can consider CD14+ cells as dendritic cells (doi: 10.1038/jid.2008.56 doi: 10.1016/j.immuni.2014.08.006, etc.). A proper discussion is required to make any conclusion in such a manner. Even in papers referenced in this paragraph, it is debated since such cells share phenotype and morphology between monocytes/macrophages and conventional dendritic cells.

11. Lines 181-185, "Recently, our group identified and characterized a subset of IL-10-producing tolDCs, named DC-10, which express CD14, CD16, CD141 and CD163. Ex vivo isolated DC-10 have a unique cytokine production profile with high ratio of IL-10/IL-12 production, co-express high levels of the tolerogenic molecules HLA-G and ILT4 and promote allo-specific Tr1 cells in vitro [48]."

In the pointed article so-called "monocyte-derived dendritic cells", which are monocytes differentiated to dendritic cell-like cells in vitro, were used. Such cells do not represent conventional dendritic cells, cannot be found physiologically, and can be hardly attributed to dendritic cells.

12. The overall impression on paragraph 2 is that there is no proper description of what tolerogenic dendritic cell is. There are only excerpts from different articles describing which immune-tolerogenic functions antigen-presenting cells can possess. What cell subset(s) do mentioned tolerogenic dendritic cells represent? What is required for antigen-presenting (dendritic) cells to become tolerogenic? Do tolerogenic dendritic cells represent the separate subsets of dendritic cells or are they regular dendritic cells that possess tolerogenic phenotype? I have not found any answers to these questions in the present manuscript.

13. Lines 227-232, “The active form of Vitamin-D3 (VitD3), 1,25-dihydroxyvitamin D3 (1,25(OH)2D3), impairs DC differentiation and maturation in vitro and in vivo, leading to a tolerogenic phenotype characterized by low Ag presentation capacity, down-regulation of costimulatory molecules, and inhibition of IL-12 secretion. VitD3-generated DCs (VitD3-DCs) are thus unable to fully activate T cells and initiate an immune response [56].”:

The reference should support the statement, but not the conclusion made from the statement.

14. Lines 263-265, “All these works demonstrated that tolDCs can be efficiently differentiated in vitro and, independently from the differentiation protocol used, there is an overall consensus regarding the phenotype and function of those cells.”:

Unfortunately, similar to what is expressed in comment 12, I do not see any consensus opinion from the tolerogenic dendritic cell description or production methods given in the present manuscript. It is either should be given in a short form with adequate references to check current opinions on what tolerogenic dendritic cells are or, if decided to represent it in detail in the present manuscript, since one-third of the entire manuscript is an attempt to describe tolerogenic dendritic cells, proper description with proper conclusions should be given.

15. Line 310, “In another study, autologous tolDCs tolerized with Dexa…”:

“Tolerogenic dendritic cells that were tolerized” sounds like a tautology.

16. Small descriptions for all diseases in paragraph 3 were given, except for rheumatoid arthritis.

17. Lines 390-391, “In the last decades, evidence demonstrated that NPs could be used also as an alternative route of administration to deliver soluble immunosuppressive drugs.”:

Nanoparticles are not a route of administration, but a way of targeting and delivery. Routes of administration are intradermal, intranodal, etc.

18. Lines 389-393, “Nanoparticles (NPs) were widely used in medicine to delivery drugs during the years. In the last decades, evidence demonstrated that NPs could be used also as an alternative route of administration to deliver soluble immunosuppressive drugs. However, this kind of drugs are not specific for the target organ and they induce a general immune susceptibility to secondary infections and malignancies in the treated patients [87]..”:

Contrariwise, the nanoparticle approach can be utilized for cell-specific substance delivery. Given reference is all about it as well. Overall, this fragment (lines 390-397) requires extensive English editing for proper understanding.

19. Lines 395-397, “The combined delivery of tolerogenic agents and Ags into the NPs is crucial to improve Ag-specific tolDC ability [88] (Figure 2).”:

The given reference provides a rationale for nanoparticle functionalization with dendritic cell-specific antibodies and showed that it is possible to deliver rapamycin and OVA peptides in such a manner to dendritic cells, but do not confirm that “delivery of tolerogenic agents and antigens within nanoparticles is crucial to improve antigen-specific tolerogenic dendritic cell ability”. Besides, what is “antigen-specific tolerogenic dendritic cell ability”?

Moreover, it is in controversy with the statement expressed further in the manuscript: lines 449-451, “However, the combination between NPs and others immunomodulatory treatments needs further investigation to verify if the synergic tolerogenic activity is improved compared to the single treatments alone.”

20. Lines 398-401, “In the context of T1D, Lewis et al. demonstrated that Ag-specific tolDCs are able to phagocyte NPs encapsulated with 1040–55 mimotope peptide and elicit an Ag-specific T cell response in BDC2.5 NOD transgenic mice.”:

It is not clear what the Authors meant by antigen-specific tolerogenic dendritic cells. Was it dendritic cells that were already pretreated with some antigens prior to nanoparticle treatment? Does it mean that before nanoparticle treatment dendritic cells were already mature (tolerogenic or not)? Otherwise, it is unclear what that “Ag-specific tolDCs” means.

21. Lines 432-433, “A potential tolDC-based Ag-specific strategy could tapper gliadin specific pathogenic T cells and restore the immunological balance.”:

The meaning of the word “tapper” in this sentence is not clear.

22. Lines 478-481, “In the context of EAE, different studies indicated that the treatment with anti-DEC-205 chimeric Abs fused with MOG35–55 or with PLP139–151 resulted in amelioration of the disease score, prevention of pathogenic T cell accumulation in the CNS, induction of anergy in T effector cells, and reduction in IL-17 secretion [101].”:

Which studies? I see only one reference to one research article.

23. Line 590: Either the word “tumor” or “tumour” should be used within the whole article.

24. Figure 1 is not representative. Such material is better to represent in a form of a table.

Author Response

Tolerogenic dendritic cell-based approaches in autoimmunity

Manuscript ID: ijms-1269544

Dear Editor,

Dear Referees,

Thank you for giving us the opportunity to resubmit our revised manuscript for publication in IJMS. We thank the reviewers for the comments. We have modified the paper and inserted the suggestions made by the referees. In the following you find an itemized response to all points raised by the reviewers.

Passeri Laura and Gregori Silvia

List of changes according to comments by referee 3

In the present review Passeri et al. describe dendritic cells as regulators of immune response (briefly, but it represents one-third of overall text), and tolerogenic dendritic cells as a tool to struggle against autoimmune diseases. The topic of a review is of current interest.

However, for a review, there are quite a lot of references to other reviews. There are statements that should be better supported by experimental article references, but not reviews, and statements that can be supported by review references, but that are without references at all. At a glance, some plagiarism could be visible with no correct references given (as one example, if you will compare lines 106-108 and the following article: DOI: 10.1016/j.it.2020.11.001, beginning of the paragraph "Immunogenesis versus Tolerogenesis: A Role for Dendritic Cells in Peripheral Tolerance").

Response: We extensively revised the manuscript by including original articles supporting our statements. However, some of the statements are generally accepted in the field of immunological tolerance. We emended the manuscript to limit possible repetitions.

Little originality can be found and it is not clear how this review can contribute to the field, except for descriptive values of what already was published in other reviews, or "references list" value. Considering all the mistakes, missing references, confusion due to language, definitions, and in general, the current manuscript seems very immature and requires at least major revision.

Response: Although we appreciate the Reviewer’ s comment, we believe that this manuscript is not just a repetition of other Reviews in the field. As stated above we included, when needed, the original research articles.

Below the most visible critical notes are listed.

  1. The following statements should be supported with references:
    • Lines 31-32, “Organ destruction in AIDs is associated to the dysregulation of the immune system, leading to chronic inflammation and hyperactivation of adaptive immune responses.”

Response: We added some original manuscripts describing that in autoimmune diseases a dysregulation in the lymphoid and myeloid cells has been associated with tissue inflammation and activation of the immune system.

  • Lines 68-70, "Once they completed maturation in the BM, pDCs migrate into the bloodstream and are characterized by the ability to secrete large amounts of type I IFNs in response to viral infection."

Response: Although it is generally accepted in the field that pDCs are the main producers of type I IFNs, we added seminal works demonstrating that upon viral activation pDCs secrete IFNa.

  • Lines 112-116, "The encountering of the Ag drives the maturation of the iDCs that lose the ability to process new peptides and acquire the capacity to present Ags to T cells. Specifically, they upregulate the expression of MHC and co-stimulatory molecules (i.e., CD40, CD80, and CD86) and secrete pro-inflammatory cytokines (i.e., IL-1β, IL-12, IL-6, and tumor-necrosis factor α (TNFα))."

Response: Although we understand the Reviewer, this is one of the characteristics accepted from the community. Anyway, we added as reference a book charter and a review summarizing the features of DCs.

  • Lines 129-133, "However, this mechanism does not fully assure the selection of T cell unresponsive to self and innocuous foreign Ags since: (i) self-reactive lymphocytes can escape negative selection, (ii) many innocuous environmental Ags, including commensal microbiota, are not expressed in the thymus, and (iii) TCR specific for foreign Ags can recognize MHC-self-Ag complexes."

Response: We understand the Reviewer’s comment, but we do not think it is necessary to add a reference. In all immunology books it is present a chapter describing thymic selection and these mechanisms.

  • Lines 139-141, "Four main mechanisms of peripheral tolerance have been described: induction of clonal anergy, metabolic modulation, secretion of antiinflammatory cytokines, and clonal deletion."

Response: As above, this is really a general statement regarding the mechanisms of peripheral tolerance, it is described in all the immunology books. We have not included specific references for this sentence.

  • Lines 149-150, "T cell anergy can be also induced by ... the induction of indoleamine 2,3-dioxygenase (IDO)."

Response: As requested, we included the reference describing the first demonstration that DCs can produce IDO, which inhibits T cell proliferation. Moreover, we included a manuscript demonstrating that in vivo administration of IDO-overexpressing cells lead to down-regulation of allo-speficic T cell responses after in vitro restimulation. This is an example of anergy induction. According to these evidences we modified the sentence above as follow: “T cell unresponsiveness and anergy can be also induced by…the induction of indoleamine 2,3-dioxygenase (IDO)”

  • Lines 158-161, "In addition to the above mechanisms, tolDCs by secreting immunomodulatory mediators as anti-inflammatory cytokines (i.e., IL-10, TGF-β, and IL-35) and metabolites (i.e., retinoic acid (RA)) are involved in promoting Treg differentiation."

Response: We apologized for the oversight, retinoic acid (RA) has been used to modulate DC and it is not secreted by DCs. Accordingly, we modified the sentence as follow: “In addition to the above mechanisms, tolDCs by secreting immunomodulatory mediators as anti-inflammatory cytokines (i.e., IL-10, TGF-β, and IL-35) are involved in promoting Treg differentiation."

  • Lines 162-164, "IL-35 derived from tolDCs promotes the differentiation of IL-35producing FOXP3+ Tregs and suppressed Th17 cell induction."

Response: We included references sustain the statement that has been modified as follow “IL-35, which can be secreted by DC (Dixon K.O. et al., EJI 2015) promotes the differentiation of IL-35-producing FOXP3+ Tregs and suppressed Th17 cell induction”, and we included the original reference describing the role of IL-35 in expanding Tregs and suppressing Th17 cells (Niedbala W. et al, EJI 2007).

  • Lines 237-239, “Addition of RAPA during DC differentiation generates cells 237

(RAPA-DCs) with low levels of co-stimulatory molecules (CD86, CD40), high CCR7 ex-238 pression, and a decrease PDL-1 expression.”

  • Lines 239-240, “RAPA-DCs induce T-cell hypo-responsive-239 ness and expand Foxp3+ Tregs in vitro.”

Response: As requested, we eliminated the Review describing the characteristics of RAPA-DC and we included the original references for RAPA-DCs.

  • Lines 421-427, “In the same autoimmune setting, IL-10/MOG encapsulating NPs injected subcutaneously in mice prior the induction of EAE showed significant inhibition of EAE development compared to controls. Splenocytes of treated mice, collected at the peak of the diseases, showed a significant impairment in the secretion of IL-17 and IFN compared to splenocyte of control mice. Histological analysis on the spinal cord of the treated mice revealed that CD3+ infiltrating T cells were increased in control group compared to mice treated with IL-10/MOG NPs.”

Response: As requested we included the missing reference.

Lines 428-432, “Celiac Disease (CeD) is an autoimmune disorder of the small intestine, caused by the exposure to dietary gluten in genetically susceptible individuals, expressing HLA-DQ2 or HLA-DQ8. The gluten peptides, of which gliadin is the immunodominant one, upon entering intestinal lamina propria are recognized as immunogenic and triggers an adaptive immune response.”

Response: We added a Review in which the pathogenesis of CeD is deeply summarized.

  • Lines 469-471, “This conjugate has been demonstrated to be efficient in the delivering the Ag. However, it showed a lower target specificity compare to the recombinant chimeric Ab, which present some modification that enhanced their specificity.”

Response: We revised the sentences and included the relevant references on the three different system described.

  • Lines 586-588, “In 2016 the CAR technology has been also applied to monocytes leading to the generation of cells able to interact with tumour Ags, allowing the selective homing of engineered cells to tumour.”

Response: We included in the revised version the original article describing the generation of CAR macrophages.

  1. Line 22: The title of the paragraph does not match the content of the paragraph. It is stated about "breakdown of tolerance" and "dysregulation of the immune system", but the reasons why it is dysfunctional are not discussed. I would suggest either add an appropriate brief discussion on why is it considered dysfunctional, or change the title.

Response: We modified the title of the paragraph as suggested and we better introduced the presence in autoimmune settings of hype-reactive immune response, which is not controlled by the tolerogenic arm of the immune system.

  1. Line 27, “(Goodnow et al., 2005).”: Reference is given in a different from the main list style.

Response: Sorry for the inaccuracy, we modified it.

  1. Lines 62-65, 99-102: It is quite controversial if so-called monocyte-derived dendritic cells can be classified as dendritic cells since they share the phenotype of both dendritic cells and monocytes/macrophages. Moreover, the term "monocytederived dendritic cells" is attributed to artificially obtained cells after induction of monocytes with IL-4 and GM-CSF in vitro, which cannot be obtained physiologically. The term "macrophages-DC progenitor" is not clear as well (line 59). Probably "monocytes-dendritic cells progenitor" was meant? Also, following the article referenced in the current manuscript (10.1038/nri3712), monocyte-derived cells are not descendants of a common dendritic cell precursor. Overall, the whole paragraph (lines 55-65) is suggested to be revised, with more references to be added.

Response: We understand that a better description of the progenitors and of the nomenclature was required. However, the origin of DCs is not the main focused of this review. According to Reviewer #2 suggestions we eliminated the paragraph.

  1. Lines 54, 103: The numbers of paragraphs are the same.

Response: We corrected them.

  1. Lines 146-147: "…PDL-1 and PDL-2 programmed death 1 (PD-1) interaction [33]…":

The meaning of this sentence is unclear, revision is required. Also, the reference seems misleading or out of context.

Response: We modified the sentence as follow: “by the engagement of inhibitory receptors with their ligands expressed on the T cells these include: PDL-1 and PDL-2 interaction with programmed death 1 (PD-1)….”.

  1. Lines 147-148: "...ILTs HLA class I molecules interaction...":

The same as in the previous note - interaction between what and what? There is no hyphens, no prepositions. Language editing is required, otherwise, it just looks like a set of words with no logic.

Response: We apologize for the misleading sentence. ILTs interact with HLA-class I molecules, thus we modified the sentence as follow: “the interaction between ILTs and classical and non-classical HLA class I molecules”.

  1. While describing the effects of tolerogenic dendritic cells reported in experiments it is important to add which cell subtypes and their species origin were used to perform experiments in referenced articles.

Response: Bone-marrow derived DCs have been used in pre-clinical models which in experiments with human cells, patients’ monocyte-derived DCs have been always used. Nevertheless, we revised carefully the manuscript and indicated when murine bone-marrow derived DCs or human monocyte-derived DCs have been used for the indicated experiments.

  1. Line 164-166, "In a preclinical model of MS, retinoic acid (RA), a metabolite of vitamin A has been used to induced Tregs and inhibit Th17 cell polarization [41].":

It is clearly written in the present manuscript and was done in the referenced paper that artificial treatment with retinoic acid induced Tregs and inhibited Th17 polarization, but not retinoic acid-secreting dendritic cells. Considering that no reference was given earlier to support that dendritic cells can secrete retinoic acid (see comment 1), it can't be claimed that dendritic cells can act through such a mechanism on T cells.

Response: We apologize for the confusion, as already indicated above we did a mistake, indeed DCs do not produce RA, but DCs differentiated in the presence of RA become tolerogenic. We modified the manuscript accordingly.

  1. Lines 168-195: It is a debatable question whether dendritic cells can express CD14, or if we can consider CD14+ cells as dendritic cells (doi: 10.1038/jid.2008.56 doi: 10.1016/j.immuni.2014.08.006, etc.). A proper discussion is required to make any conclusion in such a manner. Even in papers referenced in this paragraph, it is debated since such cells share phenotype and morphology between monocytes/macrophages and conventional dendritic cells.

Response: We respectfully disagree with the Reviewer, indeed a subset of CD141+ DCs expressing CD14 has been identified in the dermis of human skin. These cells constitutively secrete IL-10 and act as immunoregulatory tissue-resident DCs (Chu CC. et al. JEM 2012). The existence of CD14+ DCs have been also reviewed in Collin M. et al. Immunology 2013. Our group recently identified a subset of DCs expressing CD14+ in the peripheral blood (Comi et al. CMI 2020) and in gut mucosa (unpublished data). These CD14+ cells are DCs since they prime naïve CD4+ T cells. We have included a statement indicating that these cells are DCs since they prime naïve T cell responses.

  1. Lines 181-185, "Recently, our group identified and characterized a subset of IL-10producing tolDCs, named DC-10, which express CD14, CD16, CD141 and CD163. Ex vivo isolated DC-10 have a unique cytokine production profile with high ratio of IL-10/IL12 production, co-express high levels of the tolerogenic molecules HLA-G and ILT4 and promote allo-specific Tr1 cells in vitro [48]."

In the pointed article so-called "monocyte-derived dendritic cells", which are monocytes differentiated to dendritic cell-like cells in vitro, were used. Such cells do not represent conventional dendritic cells, cannot be found physiologically, and can be hardly attributed to dendritic cells.

Response: We respectfully disagree with the Reviewer, indeed we recently demonstrated that CD14+CD16+CD163+CD141+ cells can be isolated from peripheral blood (Comi et al. 2020).

The overall impression on paragraph 2 is that there is no proper description of what tolerogenic dendritic cell is. There are only excerpts from different articles describing which immune-tolerogenic functions antigen-presenting cells can possess. What cell subset(s) do mentioned tolerogenic dendritic cells represent? What is required for antigen-presenting (dendritic) cells to become tolerogenic? Do tolerogenic dendritic cells represent the separate subsets of dendritic cells or are they regular dendritic cells that possess tolerogenic phenotype? I have not found any answers to these questions in the present manuscript.

Response: Several reports reported in vivo DC with tolerogenic properties: i) DCs at immature stage involved in maintaining tissue homeostasis, preventing the activation of pathogenic responses, and maintaining tolerance, and ii) specialized subsets of DCs, characterized by specific biomarker (one example DCs in the skin) involved in promoting tissue-specific tolerance.

  1. Lines 227-232, “The active form of Vitamin-D3 (VitD3), 1,25-dihydroxyvitamin D3 (1,25(OH)2D3), impairs DC differentiation and maturation in vitro and in vivo, leading to a tolerogenic phenotype characterized by low Ag presentation capacity, down-regulation of costimulatory molecules, and inhibition of IL-12 secretion. VitD3-generated DCs (VitD3DCs) are thus unable to fully activate T cells and initiate an immune response [56].”:

The reference should support the statement, but not the conclusion made from the statement.

Response: We modified the manuscript as indicated

  1. Lines 263-265, “All these works demonstrated that tolDCs can be efficiently differentiated in vitro and, independently from the differentiation protocol used, there is an overall consensus regarding the phenotype and function of those cells.”:

Unfortunately, similar to what is expressed in comment 12, I do not see any consensus opinion from the tolerogenic dendritic cell description or production methods given in the present manuscript. It is either should be given in a short form with adequate references to check current opinions on what tolerogenic dendritic cells are or, if decided to represent it in detail in the present manuscript, since one-third of the entire manuscript is an attempt to describe tolerogenic dendritic cells, proper description with proper conclusions should be given.

Response: We respectfully disagree with the Reviewer, several investigators in the field support this conclusion. Indeed, as also pointed out by the Reviewer on the top several recent Reviews have been published on the topic.

Line 310, “In another study, autologous tolDCs tolerized with Dexa…”:

“Tolerogenic dendritic cells that were tolerized” sounds like a tautology.

Response: We modified it.

  1. Small descriptions for all diseases in paragraph 3 were given, except for rheumatoid arthritis.

Response: We included the brief description of RA.

  1. Lines 390-391, “In the last decades, evidence demonstrated that NPs could be used also as an alternative route of administration to deliver soluble immunosuppressive drugs.”:

Nanoparticles are not a route of administration, but a way of targeting and delivery. Routes of administration are intradermal, intranodal, etc.

Response: Sorry, we modified it.

  1. Lines 389-393, “Nanoparticles (NPs) were widely used in medicine to delivery drugs during the years. In the last decades, evidence demonstrated that NPs could be used also as an alternative route of administration to deliver soluble immunosuppressive drugs. However, this kind of drugs are not specific for the target organ and they induce a general immune susceptibility to secondary infections and malignancies in the treated patients [87]..”:

Contrariwise, the nanoparticle approach can be utilized for cell-specific substance delivery. Given reference is all about it as well. Overall, this fragment (lines 390397) requires extensive English editing for proper understanding.

Response: We revised the sentence as follow: ‘’In the last decades, evidence demonstrated that Nanoparticles (NPs) could be used as an innovative tool to deliver drugs or molecules to specific cell subsets.’’

  1. Lines 395-397, “The combined delivery of tolerogenic agents and Ags into the NPs is crucial to improve Ag-specific tolDC ability [88] (Figure 2).”:

The given reference provides a rationale for nanoparticle functionalization with dendritic cell-specific antibodies and showed that it is possible to deliver rapamycin and OVA peptides in such a manner to dendritic cells, but do not confirm that “delivery of tolerogenic agents and antigens within nanoparticles is crucial to improve antigen-specific tolerogenic dendritic cell ability”. Besides, what is “antigen-specific tolerogenic dendritic cell ability”?

Moreover, it is in controversy with the statement expressed further in the manuscript: lines 449-451, “However, the combination between NPs and others immunomodulatory treatments needs further investigation to verify if the synergic tolerogenic activity is improved compared to the single treatments alone.”

Response: We modified the sentence as follow: “further investigation are needed to define the best immunomodulatory molecules to be encapsulated in NPs

  1. Lines 398-401, “In the context of T1D, Lewis et al. demonstrated that Ag-specific tolDCs are able to phagocyte NPs encapsulated with 1040–55 mimotope peptide and elicit an Ag-specific T cell response in BDC2.5 NOD transgenic mice.”:

It is not clear what the Authors meant by antigen-specific tolerogenic dendritic cells. Was it dendritic cells that were already pretreated with some antigens prior to nanoparticle treatment? Does it mean that before nanoparticle treatment dendritic cells were already mature (tolerogenic or not)? Otherwise, it is unclear what that “Ag-specific tolDCs” means.

Response: The Reviewer realizes what we meant, indeed DCs can become antigen-specific after up-take NPs carrying antigens. We modified the sentence as follow: “the context of T1D, Lewis et al. demonstrated that tolDCs become able to present Ag delivered by NPs encapsulated with 1040–55 mimotope peptide and elicit an Ag-specific T cell response in BDC2.5 NOD transgenic mice”.

  1. Lines 432-433, “A potential tolDC-based Ag-specific strategy could tapper gliadin specific pathogenic T cells and restore the immunological balance.”:

The meaning of the word “tapper” in this sentence is not clear.

Response: We modified the term following the Reviewer suggestion.

  1. Lines 478-481, “In the context of EAE, different studies indicated that the treatment with anti-DEC-205 chimeric Abs fused with MOG35–55 or with PLP139–151 resulted in amelioration of the disease score, prevention of pathogenic T cell accumulation in the CNS, induction of anergy in T effector cells, and reduction in IL-17 secretion [101].”:

Which studies? I see only one reference to one research article.

Response: We added additional refs.

  1. Line 590: Either the word “tumor” or “tumour” should be used within the whole article.

Response: We modified the manuscript accordingly.

  1. Figure 1 is not representative. Such material is better to represent in a form of a table.

Response: We respectfully disagree with the Reviewer, and we maintained Figure 1, but we also included in the revised manuscript a summary table (Table 1)  with the main characteristics of human DC subsets.

Round 2

Reviewer 2 Report

The authors have made extensive modifications and revisions to the manuscript resulting in a much more readible version.  The newly inserted Table defining DC subtypes is quite clear and represents a very nice up-to-date reference. Overall, this new version of the manuscript is much easier to read and understand, plus shorter making it more likely that the reader will remain connected.

One essential change ...... line 408 ..... please correct spelling of Crohn´s. 

Author Response

We thank the reviewer for the positive assessment, we modified  the manuscript as suggested.